# Erg4 Is Involved in Ergosterol Biosynthesis, Conidiation and Stress Response in *Penicillium expansum*

**DOI:** 10.3390/jof9050568

**Published:** 2023-05-13

**Authors:** Zhanhong Han, Yuanyuan Zong, Xuemei Zhang, Di Gong, Bin Wang, Dov Prusky, Edward Sionov, Huali Xue, Yang Bi

**Affiliations:** 1College of Food Science and Engineering, Gansu Agricultural University, Lanzhou 730070, China; hanzhanhong009@163.com (Z.H.); zongyy@gsau.edu.cn (Y.Z.); gongdi531@163.com (D.G.); wangbin_1519@163.com (B.W.); dovprusk@volcani.agri.gov.il (D.P.); 2Department of Postharvest Science of Fresh Produce, Agricultural Research Organization, Volcani Center, Rishon LeZion 50250, Israel; 3Department of Food Science, Agricultural Research Organization, Volcani Center, Rishon LeZion 50250, Israel; edwardsio@volcani.agri.gov.il; 4College of Science, Gansu Agricultural University, Lanzhou 730070, China

**Keywords:** *Penicillium expansum*, *erg4s*, ergosterol, growth and development, pathogenicity

## Abstract

*erg4* is a key gene for ergosterol biosynthesis in filamentous fungi, but its function in *Penicillium expansum* remains unknown. Our results showed that *P. expansum* contains three *erg4* genes, including *erg4A*, *erg4B* and *erg4C*. The expression levels of the three genes showed differences in the wild-type (WT) strain, and the expression level of *erg4B* was the highest, followed by *erg4C*. Deletion of *erg4A*, *erg4B* or *erg4C* in the WT strain revealed functional redundancy between them. Compared to the WT strain, *erg4A*, *erg4B* or *erg4C* knockout mutants reduced ergosterol levels, with *erg4B* deletion having the greatest effect. Furthermore, deletion of the three genes reduced sporulation of the strain, and Δ*erg4B* and Δ*erg4C* mutants showed defective spore morphology. In addition, Δ*erg4B* and Δ*erg4C* mutants were found to be more sensitive to cell wall integrity and oxidative stress. However, deletion of *erg4A*, *erg4B* or *erg4C* had no significant effect on colony diameter, spore germination rate, conidiophore structure of *P. expansum* or pathogenicity to apple fruit. Taken together, *erg4A*, *erg4B* and *erg4C* have redundant functions and are all involved in ergosterol synthesis and sporulation in *P. expansum*. In addition, *erg4B* and *erg4C* contribute to spore morphogenesis, cell wall integrity and response to oxidative stress in *P. expansum*.

## 1. Introduction

*Penicillium expansum* is an important postharvest pathogenic fungus that causes blue mold in several temperate fruits. During colonization, the pathogen produces patulin and citrinin in the fruit, posing a potential threat to consumer health [1]. Ergosterol is a fungal-specific sterol found in the plasma membrane of fungi [2]. Ergosterol plays an important role in maintaining the integrity and fluidity of cell membranes and is involved in the activity of membrane proteins, transduction of signaling molecules and various biological processes [3]. A total of 20 enzymes are involved in the synthesis of ergosterol in *Saccharomyces cerevisiae* [2]. Among them, the sterol C-24 reductase, encoded by *erg4*, catalyzes the conversion of ergosta-5,7,22,24-tetraenol to ergosterol in the final step of ergosterol biosynthesis [4]. In *S. cerevisiae*, deletion of *erg4* completely blocked ergosterol biosynthesis, resulting in the accumulation of ergosta-5,7,22,24(28)-tetraenol, a precursor compound of ergosterol biosynthesis [5]. In contrast, overexpression of *erg4* promoted ergosterol production in *S. cerevisiae* [6]. Furthermore, *erg4* deletion inhibited ergosterol biosynthesis in *Xanthophyllomyces dendrorhous* [7]. The *erg4* deletion mutant of *Fusarium graminearum* showed a decrease in mycelial growth and conidiation and produced abnormal conidia as well as lower levels of DON production. Furthermore, *erg4* deletion increased the sensitivity of *F. graminearum* to osmotic and oxidative stress, but inhibited ergosterol biosynthesis and virulence against wheat heads and tomato fruits [8]. In *Aspergillus fumigatus*, *erg4A* or *erg4B* deletion had no significant effect on ergosterol synthesis and mycelial growth of the fungus. However, the mutant with both *erg4A* and *erg4B* knockout showed impaired colony growth, complete blockage of ergosterol synthesis and severe conidiation defects, but had no effect on virulence in mice [9]. Although the *erg4* gene family has been reported to regulate growth, development and pathogenicity of *S. cerevisiae* and filamentous fungi, how the *erg4* gene family affects growth, development, ergosterol biosynthesis and pathogenicity of *P. expansum* has not been reported. Our previous results showed that *P. expansum* has three homologous *erg4* genes, *erg4A*, *erg4B* and *erg4C* [10]. Transmembrane domain analysis of the corresponding encoded proteins revealed that Erg4A and Erg4C proteins contain nine transmembrane structures, while Erg4B protein contains seven transmembrane structures. In addition, subcellular localization results showed that Erg4A, Erg4B and Erg4C proteins were all localized to the endoplasmic reticulum membrane [10]. Therefore, the objectives of this study were to (1) analyze the sequence characteristics and phylogenetic relationships of Erg4A, Erg4B and Erg4C proteins; (2) construct *erg4A*, *erg4B* and *erg4C* deletion mutants and their corresponding complementation strains; (3) determine the transcription levels of *erg4A*, *erg4B* and *erg4C* and the ergosterol content in WT and mutant strains; (4) elucidate the role of the three genes in colony growth, sporulation, spore germination rate, conidiophore development and spore morphology of *P. expansum*; (5) compare the sensitivity of Δ*erg4A*, Δ*erg4B* and Δ*erg4C* mutants to osmotic stress, cell wall integrity and oxidative stress; (6) observe the pathogenicity of the three knockout mutants on apple fruit.

## 2. Materials and Methods

### 2.1. Fungal Strains, Culture Conditions and Fruit

The WT strain of *P. expansum* T01 was kindly provided by Prof. Shiping Tian, the Institute of Botany, Chinese Academy of Sciences. The WT strain and mutants used in this study were cultured on PDA medium for 7 days. Spore suspensions of each strain were collected with 10 mL of sterile water (containing 0.05% Tween-20) and then filtered through four layers of sterile gauze. A total of 100 μL of spore suspension containing 1 × 10^6^ spores/mL of each strain was inoculated into 200 mL of Czapek Yeast Extract (CY) liquid medium (containing 3 g NaNO_3_, 1 g K_2_HPO_4_·3H_2_O, 0.5 g KCl, 0.5 g MgSO_4_·7H_2_O, 0.01 g FeSO_4_·7H_2_O, 30 g sucrose, 5 g yeast extract and 1000 mL of distilled water) and incubated in a thermostatic shaker (200 rpm, 25 °C) for 3 days. The mycelia were then collected for genomic DNA extraction.

Apple fruits (*Malus domestica* Borkh. cvs. Golden Delicious and Fuji) were harvested from a commercial orchard in Jingtai county, Gansu Province, China.

### 2.2. Sequence Alignments and Phylogenetic Analysis

Amino acid sequences of Erg4 protein in different fungal species were obtained through BLASTP searches on NCBI (http://www.ncbi.nlm.nih.gov/, accessed on 15 January 2020), and multiple sequence alignments were performed using DNAMAN 6.0 software. Phylogenetic analysis was performed using MEGA 7.0 software, and the neighbor-joining (NJ) tree was constructed with a bootstrap value of 1000. The conserved motif was predicted by the online MEME program (http://meme-suite.org/, accessed on 14 March 2021).

### 2.3. Gene Knockout and Complementation

The construction of *erg4A*, *erg4B* or *erg4C* knockout strains and their complementation strains was achieved by the homologous recombination strategy (Appendix A) [11]. Briefly, the genomic DNA of the WT strain was used as a template to obtain the upstream and downstream homologous recombination sequences (approximately 1 kb) of each gene by PCR amplification using the specified primer pairs. The upstream and downstream sequences of each gene were then inserted into the pCAMBIA1300-HPH vector to obtain the corresponding knockout vector. The knockout vector of each gene was transformed into the WT strain by *Agrobacterium tumefaciens*-mediated transformation (ATMT). Transformants were selected with 250 µg/mL hygromycin B and identified by PCR amplification. Complementation strains were obtained according to the method described by [12]. The DNA fragment of *erg4A*, *erg4B* or *erg4C* was inserted into the *Xbal*I and *Sac*I site of vector pCNEO, respectively. The vector was then transformed into the corresponding mutants using the ATMT method. Transformants were selected at 250 µg/mL G-418 (Solarbio Biotechnology Co., Ltd. Beijing, China) and confirmed by PCR amplification. All primers used to generate mutants and complementation strains are listed in Appendix A.

### 2.4. Gene expression Analysis

The mycelium of *P. expansum* cultured for 3 days was collected from the CY liquid medium for the determination of gene expression. Total RNA was extracted using TRNzol Universal Reagent (Tiangen Biotech, Beijing, China) according to the manufacturer’s instructions, and then reverse transcription was performed to generate cDNA using PrimeScriptTM RT Reagent Kit with gDNA Eraser. Real-time quantitative PCR (RT-qPCR) analyses were performed using SYBR Premix Ex Taq (Takara Biotechnology Co., Ltd., Dalian, China). The *β-tubulin* gene was used as an endogenous control for normalization. Relative expression levels were calculated using the 2^−ΔΔCt^ method [13]. Primer sequences are provided in Appendix A.

### 2.5. Determination of Ergosterol Content

A 100-μL spore suspension containing 1 × 10^6^ spores/mL of either the WT, knockout mutants or complementary strains was incubated in 100 mL of CY liquid medium at 200 rpm for 3 days at 25 °C. Fresh mycelia were then collected, filtered through sterile gauze, and then washed with sterilized water three times. Approximately 200 mg of dried mycelia from each strain were treated with 3 mL of 25% alcoholic potassium hydroxide and incubated at 85 °C for 1 h. Then, 1 mL of distilled water and 3 mL of pentane were added to the mixture and vortexed for 3 min, and then kept for 10 min. The top layer was transferred to a clear tube and evaporated at room temperature in a fume hood until dry. Before analysis, all samples were dissolved in 1 mL of methanol and then filtered through a 0.22 μm filter membrane. Ergosterol concentrations were quantified using a high-performance liquid chromatography (HPLC) system (Waters, Milford, MA, USA) and detected at 282 nm [14].

### 2.6. Colony Diameter and Colony Morphology

A 2 μL spore suspension containing 1 × 10^6^ spores/mL of either the WT, knockout mutants or complementary strains was cultured on PDA medium for 7 days at 25 °C. Colony diameters were measured by the crossover method, and the colony morphology was recorded by photography [15].

### 2.7. Conidiophore Development and Spore Morphology

A 50 μL spore suspension containing 1 × 10^6^ spores/mL of either the WT, knockout mutants or complementary strains was spread evenly on PDA plates and then a sterilized coverslip was inserted into the plates at an angle of approximately 45 degrees. Hyphae were allowed to grow along the junction of the coverslip and the medium to adhere to the coverslip. After incubation at 25 °C for 1.5 days, the conidiophore structure was observed under a microscope (Olympus Corporation, Tokyo, Japan) [16].

A 2 μL spore suspension containing 1 × 10^6^ spores/mL of either the WT, knockout mutants or complementary strains was cultured on a PDA medium at 25 °C for 5 days, and then the spores were collected with sterile water and filtered through four layers of gauze to obtain a spore suspension. The spore suspension was centrifuged at 8000 rpm for 10 min, and then the supernatant was discarded. The spores were fixed in glutaraldehyde solution for 24 h, and then washed three times with PBS buffer. The spores were then dehydrated successively with different concentration gradients of aqueous ethanol (50%, 70%, 80% and 100%) for 15–20 min with each one. The samples were lightly adhered to a conductive adhesive, and then dried in a vacuum. After spraying with gold, the spore morphology was observed using a scanning electron microscope (JSM-5600LV, Tokyo, Japan) [17].

### 2.8. Determination of Sporulation and Spore Germination Rate

A 2 μL spore suspension containing 1 × 10^6^ spores/mL of either the WT, knockout mutants or complementary strains was inoculated onto PDA plates and incubated at 25 °C for 7 days. Spores were obtained by adding 10 mL of sterile water to each plate. Sporulation was counted using a hemocytometer r [18].

A 10 µL spore suspension containing 1 × 10^6^ spores/mL of either the WT, knockout mutants or complementary strains was inoculated onto PDA plates and incubated at 25 °C for 8 h. The spore germination rate of each strain was observed under a microscope (Olympus Corporation, Tokyo, Japan) [19].

### 2.9. Exogenous Stress Susceptibility Test

A 20 µL spore suspension containing 1 × 10^6^ spores/mL of either the WT, knockout mutants or complementary strains was grown on the PDA plates supplemented with 1 M NaCl, 25 mg/mL Congo red (CR) or 2 mM H_2_O_2_. After incubation at 25 °C for 7 days, colony diameters were measured by the crossover method [20].

### 2.10. Pathogenicity Test

Apple fruits were soaked in a 1% sodium hypochlorite solution for 3 min and then dried at room temperature. After surface sterilization with alcohol, three wounds (1 mm in width, 2 mm in depth) were made on the equator of each apple fruit with a sterile nail. Subsequently, a 10 µL spore suspension containing 1 × 10^6^ spores/mL of either the WT, knockout mutants or complementary strains was then inoculated into each wound. The inoculated fruits were placed in polyethylene bags and then stored at room temperature (22 ± 2 °C, RH 80–90%). After 7 days, the lesion diameter of the fruit was measured, and the lesion area was calculated according to the lesion diameter. Three replicates of each cultivar were performed, with six fruits inoculated per replicate [21].

### 2.11. Statistical Analysis

All the experiments were repeated at least three times. Excel 2020 was used to calculate means and standard errors for all data. OriginPro 2023 software (Northampton, MA, USA) was used for graphing. SPSS 26.0 software (SPSS, Inc., Chicago, IL, USA) was used to analyze the difference significance (*p* < 0.05).

## 3. Results

### 3.1. Sequence Alignment and Phylogenetic Analysis of Erg4A, Erg4B and Erg4C

The sequence alignment results showed that the amino acid sequence identity between Erg4A and Erg4B, Erg4A and Erg4C, and Erg4B and Erg4C was 21.9%, 61.7% and 19.9%, respectively (Figure 1A). The higher sequence identity of Erg4A and Erg4C proteins indicated that these two proteins may have similar biological functions in *P. expansum*.

The phylogenetic tree results showed that Erg4A, Erg4B and Erg4C proteins in *P. expansum* were separated into two branches. Among them, the Erg4A and Erg4C proteins were closely clustered, while the Erg4B protein and the Erg4 protein in *P. italicum* were located on the same branch, with the sequence identity of 97.4% (Figure 1B). These results showed that the Erg4A protein was closely related to the Erg4C protein, while the Erg4B protein was closely related to the Erg4 protein in *P. italicum*. In addition, the members of the *erg4* gene family were identified as 7 and 10 motifs, respectively. Among them, the Erg4A and Erg4C proteins contained 10 identical motifs, and the positional distribution of these motifs was uniform. The Erg4B protein contained 7 motifs (Figure 1C). These results indicated that Erg4A and Erg4C proteins may have similar functions in *P. expansum* due to their similar structure.

### 3.2. Transcript Levels of erg4A, erg4B and erg4C in WT and Mutants

In the WT strain, the expression levels of *erg4A*, *erg4B* and *erg4C* were significantly different, with the expression level of *erg4B* being the highest, followed by *erg4C*, and the lowest expression level was found in *erg4A* deletion strain. Compared to the expression of *erg4A* in the WT strain*,* the expression levels of *erg4B* and *erg4C* were 57.9-fold and 13.1-fold higher, respectively (Figure 2A). In the Δ*erg4A* mutant strain, the expression level of *erg4C* was 15.1-fold higher than that of *erg4B* (Figure 2B). In the Δ*erg4B* mutant strain, the expression levels of *erg4A* and *erg4C* were upregulated by 35.7% and 31.7%, respectively, compared to the WT. There was no significant difference between the expression levels of *erg4A* and *erg4C* in the Δ*erg4B* mutant strain (Figure 2C). In the Δ*erg4C* mutant strain, the expression level of *erg4A* was higher than that of *erg4B*, which increased by 49.5-fold (Figure 2D). These results indicate that *erg4A*, *erg4B* and *erg4C* had functional redundancy in *P. expansum*.

### 3.3. Effect of erg4A, erg4B and erg4C Deletion on Ergosterol Production

Compared with the WT strain, the ergosterol production in Δ*erg4A*, Δ*erg4B* and Δ*erg4C* was reduced by 40.6%, 48% and 27.9%, respectively, on day 7 of incubation, (Figure 3). The results indicated that the deletion of *erg4A*, *erg4B* or *erg4C* inhibited the ergosterol synthesis in *P. expansum*, with the *erg4B* knock-out having the most pronounced effect among them.

### 3.4. Effect of erg4A, erg4B and erg4C Deletion on Colony Morphology and Diameter

The colony morphology and colony diameter of Δ*erg4A*, Δ*erg4B* and Δ*erg4C* strains were not significantly different from the WT strain. The spores of the WT strain were green, whereas the color of spores of the Δ*erg4A*, Δ*erg4B* and Δ*erg4C* strains were lighter in color. The Δ*erg4A-C*, Δ*erg4B-C* and Δ*erg4C-C* strains were similar in color to the WT strain (Figure 4A,B). These results indicated that the deletion of *erg4A*, *erg4B* or *erg4C* resulted in a lighter spore color of *P. expansum* but had no apparent effect on colony morphology and colony diameter.

### 3.5. Effect of erg4A, erg4B and erg4C Deletion on Sporulation and Spore Germination Rate

Compared with the WT strain, the sporulation of Δ*erg4A*, Δ*erg4B* and Δ*erg4C* mutant strains was reduced by 40%, 44.7% and 47.1%, respectively, on day 7 of incubation. Sporulation of the Δ*erg4A-C* and Δ*erg4B-C* strains were recovered to some extent, but it was still lower than that of the WT strain. Sporulation was almost recovered in the Δ*erg4C-C* strain (Figure 5A). The spore germination rates of the Δ*erg4A*, Δ*erg4B* and Δ*erg4C* mutants and the corresponding complementary strains were not significantly different from the WT strain (Figure 5B). These results indicated that the deletion of *erg4A*, *erg4B* and *erg4C* significantly inhibited the sporulation of *P. expansum* but had no significant effect on spore germination.

### 3.6. Effect of erg4A, erg4B and erg4C Deletion on Conidiophore Development and Spore Morphology

The apical part of the sporangiophore of the WT strain expanded continuously and branched several times, producing several rounds of symmetrical or asymmetrical pedicels that produced clusters of greenish conidia at the tip. Compared with the WT strain, the sporulation structure of Δ*erg4A*, Δ*erg4B* and Δ*erg4C* strains was not significantly different, and the hyphae were able to form normal conidial heads and produce a large number of conidia (Figure 6A). The conidia of the WT strain were clustered, spherical or flattened. There was no significant difference between the conidia of Δ*erg4A* and the WT strains. However, the conidia of the Δ*erg4B* mutant showed obvious shrinkage, desiccation and water loss on the surface compared to the WT strain. A few of the Δ*erg4C* mutant conidia showed wrinkling compared to the WT strain (Figure 6B). These results indicate that among the three genes, the *erg4B* gene contributes most to the maintenance of the surface structure of *P. expansum*, followed by *erg4C*. 

### 3.7. Effect of erg4A, erg4B and erg4C Deletion on Osmotic Stress, Cell Wall Integrity and Oxidative Stress

NaCl is an exogenous osmotic pressure reagent, CR is an inhibitor of cell wall synthesis, and H_2_O_2_ is an oxidative stress pressure reagent. On the NaCl medium, there was no significant difference showed between the WT and the three knockout strains (Figure 7A). On the medium containing CR, the colony diameter of the Δ*erg4A* strain was not different from that of the WT strain, whereas the colony diameters of the Δ*erg4B* and Δ*erg4C* strains were smaller than that of the WT strain (Figure 7B). On the medium containing H_2_O_2_, no significant difference in colony diameter was found between the WT and Δ*erg4A* mutant, while the deletion of *erg4B* and *erg4C* reduced the colony diameter of *P. expansum* (Figure 7C). These results suggest that both *erg4B* and *erg4C* are involved in the responses of *P. expansum* to cell wall integrity and oxidative stress.

### 3.8. Effect of erg4A, erg4B and erg4C Deletion on Pathogenicity on Apple Fruit

Compared to the WT strain, the deletion of *erg4B* and *erg4C* slightly reduced the lesion area of Golden Delicious and Fuji fruit, but there was no significant difference between them (Figure 8). These results indicated that the deletion of *erg4A*, *erg4B* or *erg4C* had no significant effect on the pathogenicity of *P. expansum* on the apple fruits.

## 4. Discussion

In this study, a higher amino acid sequence identity was found between Erg4A and Erg4C proteins (Figure 1A). Erg4A and Erg4C proteins contained the same number of motifs (Figure 1C), indicating that Erg4A protein is closely related to Erg4C protein. Therefore, we suggest that they may play similar roles in regulating ergosterol biosynthesis in *P. expansum*. Deletion of *erg4A* induced *erg4C* expression, deletion of *erg4B* induced *erg4A* and *erg4C* expression, and deletion of *erg4C* induced *erg4A* expression (Figure 2). These results suggest that a single *erg4* deletion is compensated for by other *erg4* genes, which is similar to the results in *A. fumigatus* [9]. Therefore, we propose that *erg4A*, *erg4B* and *erg4C* have redundant roles in ergosterol biosynthesis in *P. expansum*.

The biosynthesis of ergosterol is catalyzed by a series of enzymes encoded by the *erg* gene family, using acetyl-CoA as substrate, which is a highly conserved process in fungi. [22]. In the present study, the deletion of *erg4A*, *erg4B* or *erg4C* reduced ergosterol production in *P. expansum*. Among the three *erg4* genes, *erg4B* was more important for ergosterol biosynthesis in *P. expansum*. The result was similar to that of *F. graminearum* [8]. Since *erg4A*, *erg4B* and *erg4C* are key genes involved in the final step of ergosterol synthesis [23], deletion of all three genes could inhibit ergosterol synthesis. The expression of *erg4B* in the WT strain was higher than that in the Δ*erg4A*, Δ*erg4B* and Δ*erg4C* strains (Figure 2A), suggesting that *erg4B* plays a major function in the *erg4* gene family in *P. expansum*. Therefore, we propose that *erg4A*, *erg4B* and *erg4C* are all involved in ergosterol synthesis in *P. expansum*, and that these three genes independently regulate ergosterol synthesis, of which *erg4B* has the most obvious effect.

Conidia are the main mode of asexual reproduction in filamentous fungi, and their production is tightly regulated in cells [24]. In *A. nidulans*, the *BrlA*, *AbaA*, and *WetA* genes regulate the central pathway of conidiation [25]. The *BrlA* gene modulates the transcriptional regulation of genes involved in early sporulation development; AbaA activated by BrlA is required for the differentiation process in the middle stage of sporulation, and WetA activated by AbaA is essential for spore formation and maturation [26]. It has been shown that the G protein signaling pathway plays an important role in the regulation of sporulation and hyphal growth of *A. fumigatus*. Deletion of the Gβ-like protein gene *CpcB* resulted in delayed hyphal growth and reduced sporulation of *A. fumigatus* [27]. In this study, deletions of *erg4A*, *erg4B* and *erg4C* reduced sporulation of *P. expansum* (Figure 5A), which is consistent with the results of [8], who found that the *erg4* deletion mutant formed smaller and shorter conidia with less septation and showed a reduced conidiation. In addition, we also found that the sporulation of Δ*erg4A-C* and Δ*erg4B-C* strains was lower than that of the WT (Figure 5A). We constructed the complementary strains by transferring the target gene, together with its own promoter and terminator, into the corresponding mutant strains by *Agrobacterium*-mediated transformation. In this case, the target gene was integrated into the genome of the mutant strain as a random insertion. We speculate that the gene inserted by the target gene fragment may be one that has some effect on the sporulation of *P. expansum*, resulting in the sporulation of the complementary strains not fully reverting to the WT level. Previous studies have reported that the double deletion of *erg4A* and *erg4B* completely blocked the ergosterol synthesis in *A. fumigatus* and caused severe sporulation defects, while the complementary strain completely rescued the defects, indicating that ergosterol is required for the conidiation for *A. fumigatus* [9]. Furthermore, transcriptome analysis between the WT and *erg5* knockout strains of *A. fumigatus* showed that defects in ergosterol synthesis significantly downregulated central regulatory networks (BrlA, AbaA and WetA), the genes encoding heterotrimeric G-related proteins (GpaB, RicA, GpgA, RgsA) and the MAPK (MpkC, SakA) signaling pathway [28]. However, how *erg4* regulates the expression of genes involved in these signaling pathways has not been reported yet. Based on this information, we hypothesize that *erg4A*, *erg4B* and *erg4C* deletion reduces the sporulation of *P. expansum* by decreasing ergosterol levels and down-regulating the central regulatory network as well as the heterotrimer G-related protein and MAPK signaling pathways, whereas the details of the regulatory mechanism remain to be elucidated.

The cell wall is the first barrier of filamentous fungi to resist external stresses, and it plays an important role in maintaining cell morphology [29]. CR is an inhibitor of cell wall synthesis that binds mainly to chitin and *β*-1, 4-glucan in the cell wall [30]. In this study, Δ*erg4B* and Δ*erg4C* colony growth was inhibited under CR stress (Figure 7B), which is consistent with the results in *A. fumigatus* [9]. In *S. cerevisiae*, *erg* deletion inhibited ergosterol synthesis and reduced ergosterol content, leading to increased chitin synthesis and abnormal cell wall distribution [31]. Reduced septations in conidia and mycelia were found in *F. graminearum erg4* knockout mutants, suggesting that *erg4* affects cell wall formation [8]. In *S. cerevisiae*, *erg4* has been shown to be involved in cell wall assembly [32]. Additionally, *erg4A* may also be involved in the cell wall synthesis process in *A. fumigatus*, as the Δ*erg4A* mutant shows significant sensitivity to CR, indicating that *erg4* plays an important role in fungal cell wall integrity [9]. Disruption of ergosterol synthesis could also disrupt cell wall structure [33]. Therefore, we speculate that *erg4B* and *erg4C* could disrupt the cell wall integrity in *P. expansum* by inhibiting ergosterol synthesis. Furthermore, Δ*erg4B* and Δ*erg4C* strains were sensitive to H_2_O_2_ stress (Figure 7C), which is similar to the results of Liu et al. [8] and Long et al. [9]. They found that the deletion of *erg4* in *F. graminearum* and the deletion of *erg4A* in *A. fumigatus* increased the sensitivity of the fungi to H_2_O_2_ stress [8,9]. Deletion of *erg4B* and *erg4C* reduced the levels of ergosterol, which is an important component of the fungal cell membrane. Therefore, we hypothesized that the deletion of *erg4B and erg4C* would reduce ergosterol levels in *P. expansum*, leading to an increased sensitivity of cell membranes to H_2_O_2_ stress.

Deletion of *erg4A*, *erg4B* or *erg4C* did not attenuate the pathogenicity of *P. expansum* on apple fruit (Figure 8), which is similar to the finding that the deletion of *erg4A* or/and *erg4B* did not affect the virulence of *A. fumigatus* to mice [9]. However, in contrast, the deletion of *erg4* in *F. graminearum* reduced the pathogenicity of the fungus in wheat and tomato fruits, as only one *erg4* gene is present in *F*. *graminearum* [8]. In contrast, *A. fumigatus* had two *erg4* genes (*erg4A* and *erg4B*) with functional redundancy [9]. Therefore, we considered that the deletion of *erg4A*, *erg4B* or *erg4C* did not affect the pathogenicity of *P. expansum*, which was related to the functional redundancy of the family genes. Whether the double or triple knockout of *erg4A*, *erg4B* and/or *erg4C* affects the pathogenicity of *P. expansum* requires further verification.

A global transcriptome analysis is of great importance to elucidate the function of *erg4A*, *erg4B* and *erg4C* in *P. expansum*. In addition, we will construct double knockout mutants to further elucidate the mechanism of action of these three *P. expansum* genes in regulating ergosterol biosynthesis in our future study.

## 5. Conclusions

Deletion of *erg4A*, *erg4B* or *erg4C* inhibited the ergosterol levels in *P. expansum*, whereas the *erg4B gene* had a greater effect on ergosterol biosynthesis. Deletion of *erg4A*, *erg4B* or *erg4C* also reduced the sporulation of *P. expansum*. Furthermore, *erg4B* and *erg4C* were involved in the spore morphology, cell wall integrity and oxidative stress response of *P. expansum*. However, deletion of *erg4A*, *erg4B* or *erg4C* had no significant effect on colony growth, osmotic stress of *P. expansum* or pathogenicity on apple fruit.

## Figures and Tables

**Figure 1 jof-09-00568-f001:**
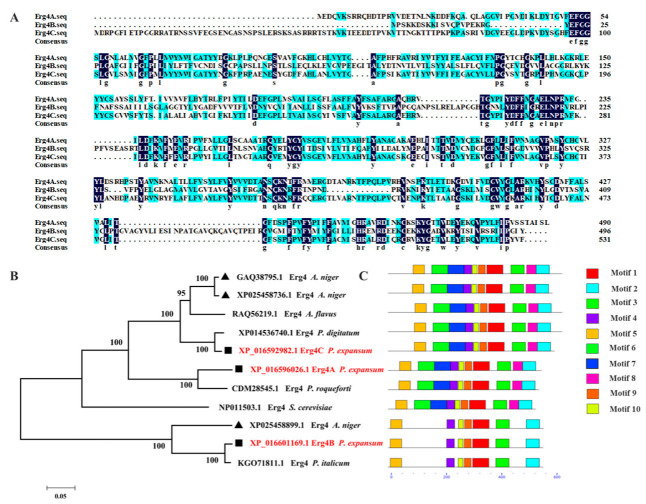
Bioinformatic analysis of Erg4A, Erg4B and Erg4C. (**A**) Alignment of amino acid sequences of Erg4A, Erg4B and Erg4C from selected fungi, including *A. niger*, *A. flavus*, *P. digitatum*, *P. expansum*, *P. roqueforti*, *P. italicum*. (**B**) The phylogenetic tree (neighbor-joining tree) was created using MEGA 7 software.(**C**) The conserved motif prediction of Erg4A, Erg4B and Erg4C by MEME program.

**Figure 2 jof-09-00568-f002:**
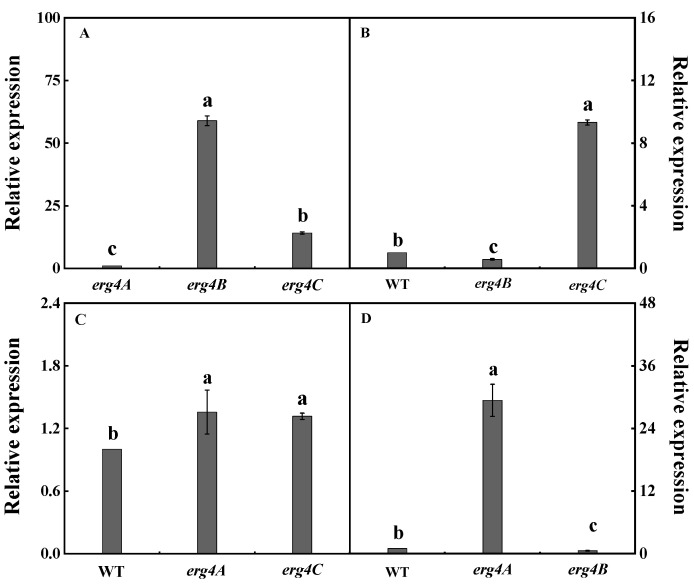
Transcript levels of *erg4A*, *erg4B* and *erg4C* in WT (**A**), and Δ*erg4A* (**B**), Δ*erg4B* (**C**) and Δ*erg4C* (**D**) mutants, respectively. Bars are the standard errors of the means. Different letters in the columns indicate significant differences (*p* < 0.05).

**Figure 3 jof-09-00568-f003:**
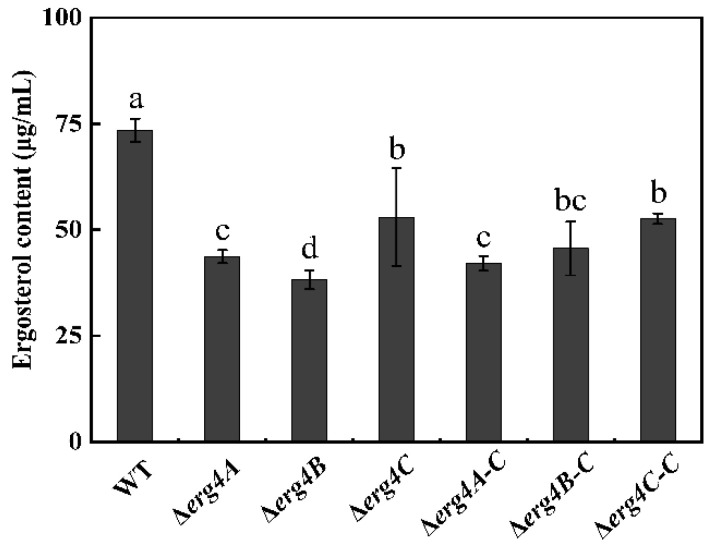
Ergosterol content of WT, *erg4* mutants and complementary strains cultured in CY liquid medium for 3 days. Bars are the standard errors of the means. Different letters in the columns indicate significant differences (*p* < 0.05).

**Figure 4 jof-09-00568-f004:**
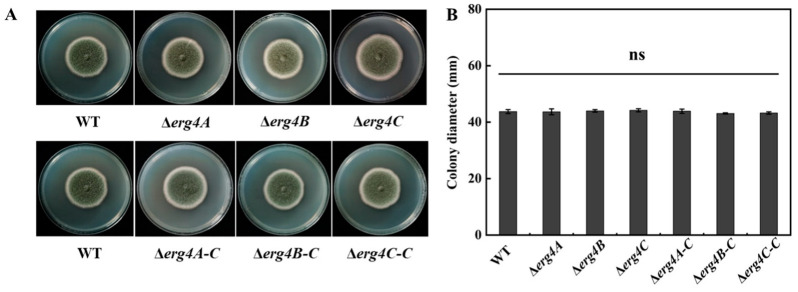
Colony morphology (**A**) and diameter (**B**) of WT, *erg4* mutants and complementary strains grown on PDA medium for 7 days. Bars are the standard errors of the means. ns indicates no significant difference (*p* < 0.05).

**Figure 5 jof-09-00568-f005:**
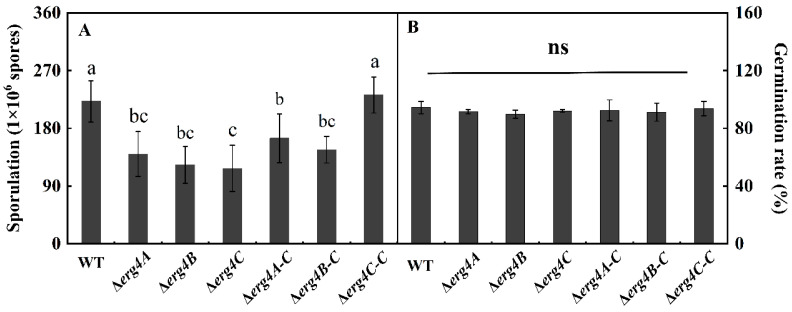
Sporulation (**A**) and spore germination rate (**B**) of WT, *erg4* mutants and complementary strains on PDA medium for 7 d and 8 h, respectively. Bars are the standard errors of the means. Different letters in the columns indicate significant differences (*p* < 0.05). ns indicates no significant difference.

**Figure 6 jof-09-00568-f006:**
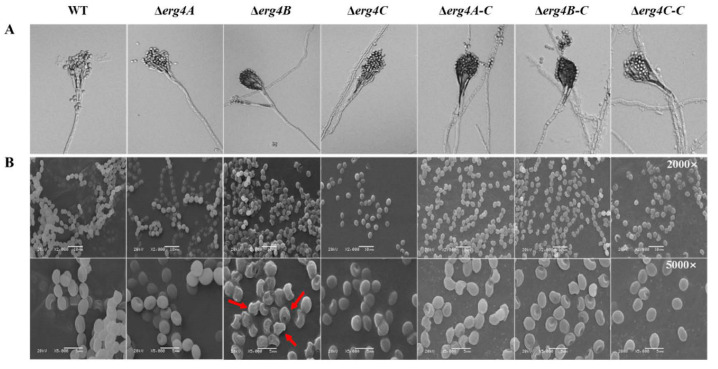
Conidiophore (**A**) and spore morphology (**B**) of WT, *erg4* mutants and complementary strains on the PDA medium for 1.5 days and 5 days, respectively. Red arrow indicates spore morphological shrinkage.

**Figure 7 jof-09-00568-f007:**
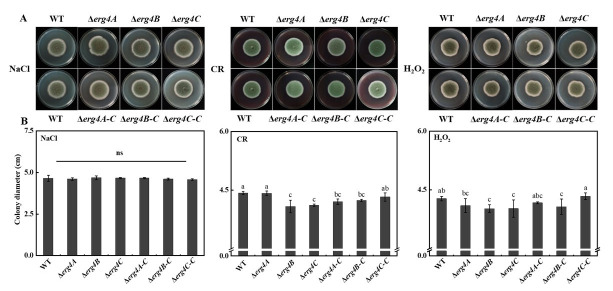
Colony morphology (**A**) and colony diameter (**B**) of WT, *erg4* mutants and complementary strains after 7 days of stress treatments. Bars are the standard errors of the means. ns indicates no significant difference. Different letters in the columns indicate significant differences (*p* < 0.05).

**Figure 8 jof-09-00568-f008:**
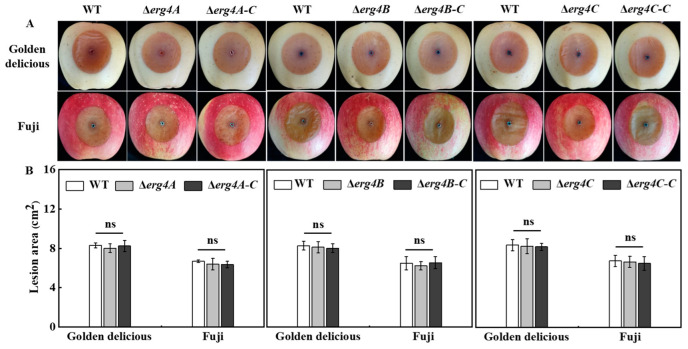
Disease symptoms (**A**) and lesion area (**B**) of WT, *erg4* mutants and complementary strains on two apple cultivars inoculated for 7 days. Bars are the standard errors of the means. ns indicates no significant difference.

## Data Availability

Not applicable.

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
