# Peer review of "Erg4 Is Involved in Ergosterol Biosynthesis, Conidiation and Stress Response in *Penicillium expansum"

_jof, 2023, doi:10.3390/jof9050568_

Round 1
Reviewer 1 Report
The research is interesting. Knockout mutants have been obtained to elucidate the function of the erg4 genes in Penicillium expansum and also complementation was used.
Since these strains have been obtained, it would have been reasonable to perform a global transcriptomic analysis. ¿Have you considered this as the next step?
None
Author Response
Reviewer 1 Comments
1. The research is interesting. Knockout mutants have been obtained to elucidate the function of the erg4 genes in Penicillium expansum and also complementation was used. Since these strains have been obtained, it would have been reasonable to perform a global transcriptomic analysis. Have you considered this as the next step?
Response: Thank you for your comments. We will follow your suggestion based on a well-designed experiment in our future study. We have added the information in the discussion section as suggested below:
A global transcriptome analysis is of great important to further elucidate the function of erg4A, erg4B and erg4C in P. expansum. Line 370-371.

Reviewer 2 Report
The manuscript titled “Erg4 involved in ergosterol biosynthesis, conidiation and stress response in Penicillium expansum” Han et al. involves the study of erg4A-C genes, key for ergosterol biosynthesis in P. expansum. Expression level of erg4B was the highest, followed by erg4C. erg4A-C present functional redundancies. Deletion of erg4A, erg4B or erg4C results in slight reduced ergosterol levels and also slight reduction of conidiation compared to WT, presenting defective spore morphology. Δerg4B and Δerg4C mutants were more sensitive to cell wall integrity tests and oxidative stress. Colony diameter, spore germination rate, conidiophore structure or pathogenicity to apple fruit were not affected by deletion of these genes. The manuscript is descriptive and contributes to the knowledge of erg4 genes in P. expansum. Please see comments below:
Title
-“Erg4 involved” for “Erg4 is involved”
Introduction:
Please revise English style
-Line 55 S. “cerevisiae and other filamentous fungi” – eliminate other
-Line 58, cite reference of “previous results” here
Results
-Line 238-239 “Sporulation of the Δerg4A-C and Δerg4B-C strains recovered to some extent, but it was still lower than that of the WT strain. S “ Could the authors provide an explanation on why the com strain is not recovering wildtype levels?
- Results with CR suggest a cell wall defect, however no cell wall analysis was performed.
-Main point: Since the erg4 genes have overlapping functions, it would be necessary to generate double and triple mutants to further understand their function.
Discussion
-Discussion presents some redundancies with respect to the Introduction section, for example in lines 292-294
- Main point: Some other regulatory genes are mentioned in the Discussion. However the manuscript does not include information on how erg4 genes could affect expression of those genes, or the expression of genes related to other processes affected by erg4 genes. Inclusion of gene expression analysis would improve this study.
Figures:
-Panels in Fig. 7A need labels
Minor improvements are needed.
Author Response
Reviewer 2 Comments
1. Title- “Erg4 involved” for “Erg4 is involved”.
Response: We have changed the title from “Erg4 involved in ergosterol biosynthesis, conidiation and stress response in Penicillium expansum” to “Erg4 is involved in ergosterol biosynthesis, conidiation and stress response in Penicillium expansum”.
2. Please revise English style.
Response: The English version of this manuscript was edited and improved by SikaPress Translation Co., Ltd in the USA. We hope that the revised version meets the requirements. Changes have been marked in blue throughout the manuscript.
3. Line 55: “S. cerevisiae and other filamentous fungi”– eliminate other.
Response: The “other” has been deleted as suggested in line 57.
4. Line 58, cite reference of “previous results” here.
Response: We have quoted a reference as suggested in line 60.
Han, Z. H.; Zong, Y. Y.; Zhang, X. M.; Wang, B.; Prusky, D.; Bi, Y. Bioinformatic, subcellular localization and expression analysis of erg4 in Penicillium expansum. Biotechnol. Bull. 2021, 37, 60–70. doi: 10.13560/j.cnki.biotech.bull.1985.2021-0141. (in Chinese with English summary).
5. Line 238-239 “Sporulation of the Δerg4A-C and Δerg4B-C strains recovered to some extent, but it was still lower than that of the WT strain. Could the authors provide an explanation on why the com strain is not recovering wildtype levels?
Response: Thank you for your comments. We have added the explanation in the discussion section as suggested below:
We constructed the complementary strains by transferring the target gene, together with its own promoter and terminator, into the corresponding mutant strains by Agrobacterium-mediated transformation. In this case, the target gene was integrated into the genome of the mutant strain as a random insertion. We speculate that the gene inserted by the target gene fragment may be one that has some effect on the sporulation of complementary strains in P. expansum, resulting in the sporulation of the complementary strains not fully returning to the WT levels. Line 321-327.
6. Results with CR suggest a cell wall defect, however no cell wall analysis was performed.
Response: We have provided more discussions on cell wall as suggested below:
In S. cerevisiae, Erg4 has been shown to be involved in cell wall assembly [32]. Additionly, Erg4A may also be involved in the cell wall synthesis process in A. fumigatus, as the Δerg4A mutant shows significant sensitivity to CR, suggesting that Erg4 plays an important role in fungal cell wall integrity [9]. Line 348-351.
Tiedje, C., Holland, D. G., Just, U., Höfken, T. Proteins involved in sterol synthesis interact with Ste20 and regulate cell polarity. J. Cell Sci. 2007, 120, 3613–3624. doi: 10.1242/jcs.009860.
7. Since the erg4 genes have overlapping functions, it would be necessary to generate double and triple mutants to further understand their function.
Response: In fact, we have constructed three double knockout vectors (erg4AB, erg4AC and erg4BC) using the same method as the construction of the single knockout vector. However, we failed to obtain a double knockout transformant after many transformation. This may be because the vector is not suitable. We have added the explanation in the discussion section as suggested below:
In addition, we’ll construct double knockout mutants to further elucidate the mechanism of action of these three P. expansum genes in regulating ergosterol biosynthesis in our future study. Line 371-373.
8. Discussion presents some redundancies with respect to the Introduction section, for example in lines 292-294.
Response: We have deleted this sentence “Sterol C-24 reductase, encoded by erg4, catalyses the final step of ergosterol biosynthesis, which converts ergosta-5, 7, 22, 24(28)-tetraenol to ergosterol” in the discussion section of revised manuscript.
9. Some other regulatory genes are mentioned in the Discussion. However the manuscript does not include information on how erg4 genes could affect expression of those genes, or the expression of genes related to other processes affected by erg4 genes. Inclusion of gene expression analysis would improve this study.
Response: We have added the information in the discussion section as suggested below:
However, how erg4 regulates the expression of genes involved in these signalling pathways has not yet been reported. Line 335-336.
10. Panels in Fig. 7A need labels
Response: We have added a corresponding label in Figure 7A.

Round 2
Reviewer 2 Report
The authors have addressed this reviewer's comments.